# Application of Physical and Numerical Modeling for Determination of Waterway Safety under the Bridge in Kaunas City, Lithuania

Tomasz Dysarz [1,*], Tomasz Kałuża [1], Karolis Mickevičius [2], Jonas Veigneris [2], Paweł Zawadzki [1], Sebastian Kujawiak [1], Stanisław Zaborowski [1], Joanna Wicher-Dysarz [1], Natalia Walczak [1], Jakub Nieć [1] and Raimundas Baublys [3]

1 Faculty of Environmental Engineering and Mechanical Engineering, Department of Hydraulic and Sanitary Engineering, Poznan University of Life Sciences, 60-637 Poznan, Poland
2 UAB "Inžinerinis Projektavimas", 03160 Vilnius, Lithuania
3 Faculty of Engineering, Department of Water Engineering, Vytautas Magnus University Agriculture Academy, 53361 Kaunas, Lithuania
* Correspondence: tomasz.dysarz@up.poznan.pl

**Abstract:** The main problem presented in this paper is the safety inlet navigation of the waterway below the bridge in the city of Kaunas in Lithuania. The analyzed reach is located in the Nemunas river downstream of the Kaunas dam. It is a part of the waterway E–41 leading to the Klaipeda harbor on the southern coast of the Baltic Sea. The work was initiated by the Lithuanian company UAB "Inžinerinis projektavimas" with funds from the project called European Union Trans-European Transport Network (EU TEN-T). The main requirement imposed along this reach is to keep sufficient depth even in the range of the lowest flows. The depth is sufficient if it is not lower than 1.15 m for minimum flows such as $Q_{95\%}$ and $Q_{95\%}$ with ice. The hydraulic conditions for maximum flow $Q_{50\%}$, $Q_{5\%}$, and $Q_{1\%}$ are also taken into account for control because the threat of hydraulic jump generation was also noticed. The research is based on georeferenced data from public and non-public sources. The hydrologic data were received from the Lithuanian Hydrometeorological Service. The physical model was created in the Water Laboratory of the Department of Hydraulic and Sanitary Engineering at Poznan University of Life Sciences, Poland. The preprocessing of spatial data in ArcGIS 10.8.2 and rules of hydraulic similarity were implemented in the process of physical model preparation. Three experiments were conducted in the laboratory with scaled values of $Q_{95\%}$, $Q_{5\%}$, and $Q_{1\%}$. The measurements of the water surface and evaluations of the average velocity were used to validate the 2D numerical model prepared in HEC-RAS 6.3.1. The basic layers of the HEC-RAS model were preprocessed in ArcGIS 10.8.2 by ESRI company. The numerical model was implemented to test different values of unknown roughness of the channel bottom. The simulations were conducted for the real values of $Q_{95\%}$ and $Q_{95\%}$ with ice and $Q_{50\%}$. The results of the simulations were depth and Froude number maps. These maps were classified into zones of no risk, middle risk, and high risk. ArcGIS in the post-processing phase was applied to identify the locations of the hazards. The magnitude of risk was expressed in terms of minimum depth achieved, maximum Froude number, as well as the length of the reaches with high risk related to these two factors. The threat of hydraulic jump formation below the bridge was also noticed. Conducted results confirmed that the combination of hydrodynamic simulations and geoprocessing in the pre- and post-processing stages could be a powerful tool in hydraulic engineering analyses. Additionally, it is worth noting that numerical modeling enables a wider analysis of potential conditions than could be possible with a physical model only.

**Keywords:** inland waterway hydraulics; river flow modeling; TEN-T program; physical modeling; geoprocessing

## 1. Introduction

In the presented research, the main problem investigated is the safety of the waterway where structures like bridges exist. Although the importance of this issue has been known for decades, a new approach is possible due to the extensive development of modern technologies. The traditional method for detailed analysis of such a problem is based on the physical modeling of the flow phenomena close to the structure of interest. However, today it is possible to apply for this purpose digital technologies like GIS processing and hydrodynamic simulation. A combination of such methods may provide more reliable results and seems to open new opportunities.

Inland waterways and waterway transport are important branches of transport in countries with a developed network of rivers and channels [1]. Several efforts have been made in the European Union aimed at the development and integration of transport networks, including inland navigation systems. During the last decade, one of the most known is the Trans-European Transport Networks (TEN–T) [2–4]. The main purpose of the TEN–T is the coordination and compatibility of investments in the transport networks for the continuity of people and goods movements in the EU states. This is an excellent opportunity for the development of waterways, especially for younger EU members located in the eastern, post-Soviet zone of the Union, like Lithuania and Poland. In both countries, there are river reaches of important international waterways. In Lithuania, the E–41 on the Nemunas river links the city of Kaunas with the Baltic Sea harbor Klaipeda. Another waterway, E–70, goes from Klaipeda through the harbor in Kaliningrad (Polish: Królewiec, Lithuanian: Karaliaucius) to the western European network of inland and marine waterways [5,6]. In 2016, the project of modernization and development of the E–41 started in Lithuania. In Poland, there are three waterways of international importance, namely E–30, E–40, and E–70. None of them have been modernized in the TEN–T project yet, but such an opportunity still exists.

There are also other important programs for the development of waterways in Europe. One of them is the convention called the European Agreement on Main Inland Waterways of International Importance (AGN), established in 1997. The countries that signed the convention are obliged to modify the parameters of their waterways to be compatible with international class IV, denoted with the capital letter E [7]. The AGN program links the waterways of European countries in the European Union (EU) and beyond, from the Atlantic coast on the west to the Ural Mountains on the east. Lithuania was one of the first countries to sign the convention. Meanwhile, Poland joined the AGN at the end of 2017 [8]. Another initiative worth mentioning is the European Inland Waterway Transport Platform [9]. The initiative was established in 2018 thanks to continuous efforts of the European Barge Union (EBU) and European Skippers Organization (ESO). Now it functions like a separate legal entity and supports projects related to the management and development of inline shipping.

Safe inland navigation requires several conditions to be preserved in the functioning and management of the waterway. The most important is proper depth, which depends on the waterway's importance. For example, there is 822 km of inland waterways of national importance to the Republic of Lithuania, and 435 km of them are under operation. According to the AGN agreement, inland waterways of the River Nemunas and the Curonian Lagoon from Kaunas to Klaipeda are inland waterways of international importance E–41 (the length is 291.2 km). The required depths for the highest four classes vary from 1.2 to 1.5 m [10]. In Poland, five classes are defined, and some of the existing classes are split into subclasses [11–13]. The depth requirements vary between 1.2 and 2.8 m. Taking into account the hydraulics of the river channels, another condition related to the flow velocity seems to also be important. This element is not considered in the majority of legal regulations due to the expectation that the navigational channels will have a relatively small slope and low velocities. This assumption may not be satisfied in the cases of larger rivers, where seasonal variability of conditions is observed. However, the problem has been observed, and it is sometimes mentioned in academic books, e.g., Kulczyk and Winter [14]. It is also considered in scientific papers related to the functioning of the waterways [15–18].

Preservation of proper hydraulic conditions is not an easy task in alluvial rivers, where intensive sediment transport is observed. The sediment deposition and removal in the processes of sedimentation and erosion may significantly change the bed profile as well as the shape of the cross-sections [18,19]. The problem becomes heavier when the structures like bridges exist and make the hydraulic conditions and navigation more complex. In such cases, a detailed analysis of potential flow conditions should be conducted.

The most traditional way of dealing with the problem of interactions between the river flow and bridges or other hydraulic structures is based on physical modeling. The aim of hydraulic model studies is to verify the functioning of designed hydroengineering investments on a smaller-scale model in relation to the actual one [20]. On the one hand, a model makes it possible to reduce or avoid potential costs if the realized investment requires retrofitting to meet the assumed conditions, while on the other hand, it facilitates the prompt introduction of changes in the dimensions and shapes of elements of a designed structure to ensure the most advantageous effect. The general principle for the physical model is to maintain the character of motion found in nature. The essence of studies using physical models is to use a mockup of the designed object to simulate water flow and the transport of debris observed in the vicinity of this structure [21]. Physical models require adequate laboratory facilities and qualified research staff, making such studies costly [22]. In general, the procedure is time and resource consuming and provides only limited opportunities. The modification of the water system requires rebuilding of the laboratory model. In many cases, the scaling is not easy and leads to some problems that are difficult to solve, like scaling of surface roughness. However, physical models are still irreplaceable in all cases, and the phenomena cannot be described analytically [23].

The development of modern technologies based on computer simulations is leading to the replacement or complement of physical modeling in many areas of engineering and physics. Mathematical models make it possible to simulate and forecast phenomena at various levels of detail [24,25]. Especially in the field of hydraulic structures and river flow interactions, methods of this kind are extremely useful. In general, mathematical modeling provides more detailed and additional information gathered from field measurements [26] and facilitates forecasts of changes in hydrodynamic conditions, e.g., resulting from climate change. The basic approaches include the implementation of hydrodynamic models based on the theory of shallow water flow [24,27]. In general, such concepts may easily be applied to analyze the problems specific to waterways [16,28,29]. In more advanced applications, complex models based on the computational fluid dynamics (CFD) theories are implemented [17,18,30]. GIS technology is also widely used in this area [31,32]. Although the numerical simulations supported with GIS provide huge opportunities, such approaches are constrained by the need for model calibration and validation. It is crucial when the design of a new waterway is tested or when measurements are too expensive or difficult. These limitations may be overcome by a combination of physical and numerical modeling as presented in Muste et al. [33].

The purpose of the presented research is to integrate physical modeling and numerical simulations supported by GIS techniques to determine the safety of inland navigation conditions in the river reach, including the bridge. The uncertainty of the roughness coefficients is carefully considered. The presented research is conducted for the reach of the Nemunas river close to the city of Kaunas in Lithuania. The physical modeling is implemented directly to validate the results provided by numerical simulations. The simulations with computer models enable an extension of the experiments made in the laboratory. The cases with different roughness of the bottom may be easily performed with a validated model. Finally, the GIS processing of the results is the basis for the assessment of safety on the basis of previously defined criteria.

## 2. The Study Object

The Nemunas River is the fourteenth largest in Europe and fourth largest in the Baltic Sea basin. It originates near Minsk (Belorussia) and drains into the Curonian Lagoon

and ultimately into the Baltic Sea. Around 940 km long, the river basin of the Nemunas stretches across four countries, namely Poland, Russia (Kaliningrad Oblast), Lithuania, and Belorussia [34]. The catchment area of the Nemunas River has a characteristic pear-like shape, which is typical for large river basins. (Figure 1a–c) The Nemunas River basin stretches from the northeast to the southwest. The catchment area is 98,200 km$^2$ (46,755 km$^2$ in Lithuania) with an average discharge of 632 m$^3$/s. The average height of the catchment area is 75 m above sea level, and the average gradient is 11.8 %. According to the general classification of rivers based on discharge and drainage basin size [35], the Nemunas River is classified as large (basin size 10 000–100 000 km$^2$, river width 200–800 m, average discharge 100–1000 m$^3$/s). Based on the classification of lowland and highland rivers [36], the Nemunas can be regarded as a pure lowland river (channel slope < 0.4‰).

The hydrographic network of the Nemunas River basin was formed in the late quaternary period. The upstream section of the basin is the oldest, formed before the last glacial period, while the midstream and downstream segments were formed during the last glacial period. The Nemunas River basin is characterized by a dense river network. From its source to its mouth, the river has about 180 tributaries [37].

In 1959, the Nemunas River near Kaunas was dammed, and the Kaunas Lagoon was formed in its flooded valley. It is the only hydropower plant on the Nemunas River in Lithuania (installed capacity P = 101 MW, head H = 20 m). The Kaunas lagoon is the largest artificial water body in Lithuania, formed in the Nemunas valley, above the Kaunas hydropower plant dam, 223.4 km from the mouth of the Nemunas. The area of the lagoon is 65.4 km$^2$, length 85 km, total length of the coast 220.3 km, and maximum depth 24.6 m. The hydropower plant causes the frequent and steady fluctuation of the water level in the Nemunas River below the dam [38–41].

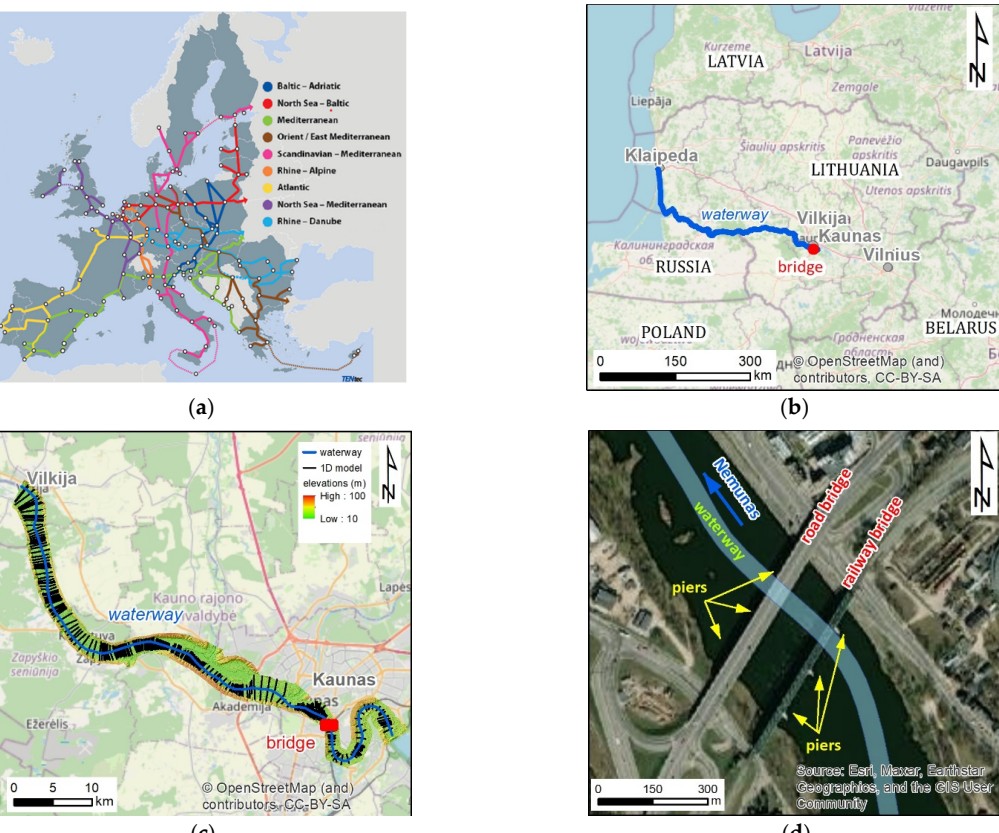

**Figure 1.** Location and elements of the investigated bridges: (**a**) TEN-T Core Network Corridors [42], (**b**) Kaunas-Klaipeda corridor and location of the bridge; (**c**) reach of the Nemunas Kaunas—Vilkija, (**d**) the bridge and its main elements.

The study object—the Kaunas Railway Bridge—was constructed in 1862. The bridge was destroyed several times. After World War II, the bridge was rebuilt. The main characteristics of the bridge are length—360 m, width—10.35 m, the construction consisting of steel trusses, steel beams, and eight spans with reinforced concrete foundations. The bridge has seven supports, three of which are in the water (Figure 1d). The railway bridge crossed the Nemunas River 11.5 km downstream from the Kaunas Hydro Power Plant (HPP).

The bridge acquired its current appearance between 1945 and 1948, when it was last rebuilt. Spans are 13.3 m long. The third span trusses with a continuous wall for each railway track; its length is 16.4 m. The trusses of the fourth and fifth spans (52.78 + 78.26 m) were made in German factories and installed during the war. The trusses of the sixth and seventh spans (77.4 + 77.4 m) are new and made of steel. The overlay of the eighth span consists of two continuous steel trusses of span length 12.94 m. Towers and frames were bricked while rebuilding the bridge on old supports in 1946. The 4th, 5th and 6th towers, the water, have German-made monolithic, reinforced caisson foundations and concreted grating (the upper part of the foundation, the pile). As before, two railway lines were built over the bridge [43].

The bridge is an important part of the historical heritage of Kaunas, so it is given increased attention. The bridge is part of the currently being built Rail Baltica. (Figure 2).

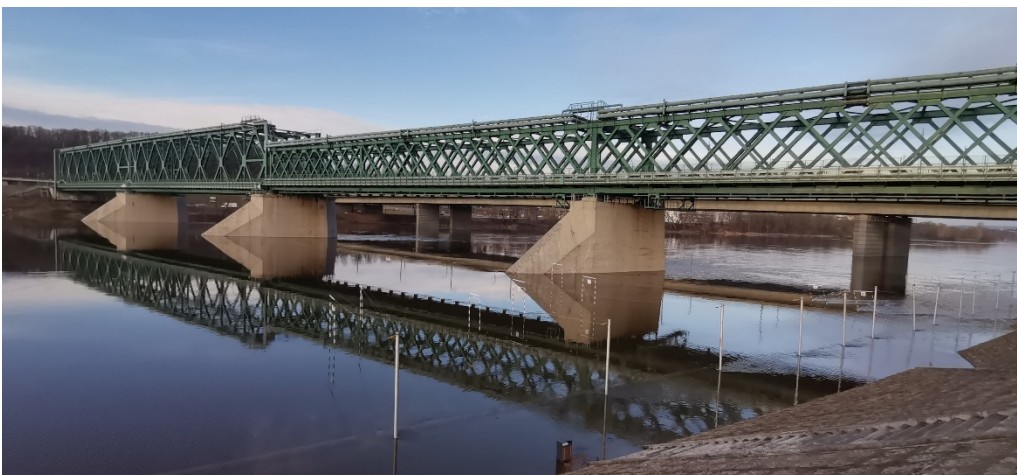

**Figure 2.** Study object: Kaunas Railway Bridge (source: R. Baublys).

### 3. Materials and Methods

Five types of data are the basis for the presented research. These are (1) the digital terrain model (DTM), (2) bathymetric measurements, (3) design of the bridge in CAD format, (4) hydrologic information about potential flows, and (5) information about the bottom cover. These data are completed with information about waterway safety criteria in Lithuania based on the requirements of the depth in low flow conditions. The additional safety criteria for the Froude number were also specified, though there are no official documents referring to such conditions according to the author's knowledge. The spatial data also include layers used for visualization.

The first two types of data are used in the GIS formats. The digital terrain model of the riverbed and surrounding areas was provided by the Lithuanian company UAB"Inžinerinis projektavimas". The DTM was prepared for the purpose of the EU Flood Directive [44] implementation in the territory of Lithuania between 2011–2012. The raster is stored in GRID format with the resolution 1 m x 1 m. The DTM covers the areas where the flood hazard and flood risk were determined in Lithuania according to the requirements of the EU Flood Directive. It encompasses all the greater rivers with their valleys including also important inundation areas. However, the part important for this research is smaller. As it is shown in Figure 3a, it consists of the riverbed under the bridge with short sections upstream and downstream, plus the banks and part of the floodplains.

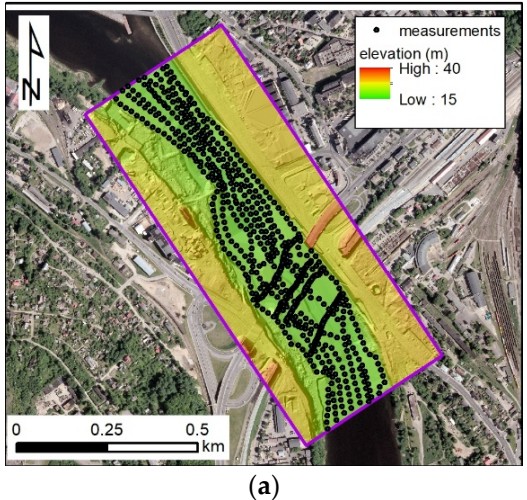
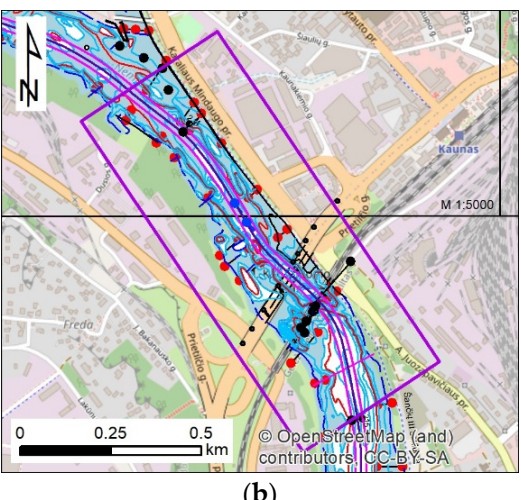

(**a**)  (**b**)

**Figure 3.** The maps and design of the investigated bridge: (**a**) ortophotomap with digital terrain model (DTM) of the bridge and measurements of the bathymetry, (**b**) design in CAD format converted to ArcGIS with OpenStreetMap in the background.

In the same figure (Figure 3a), the second type of data, the bathymetry measurements, are presented. These measurements were done in 2017 and provided by the UAB"Inžinerinis projektavimas". The device implemented is a single frequency echosounder Echologger EU 400 (EofE Ultrasonics Co., Ltd, Gyungki-Do, Korea). Its horizontal and vertical accuracies are about 1 cm [45]. The measurement points were used for the reconstruction of riverbed bathymetry at the time of the research, after several years from the previous measurements done for the EU Flood Directive implementation. It was necessary due to observations of intensive sediment transport in the Nemunas river, as well as accumulation and erosion areas near the investigated reach.

The geometry of the analyzed object and modeled environment is completed with the design of the bridge in CAD format (Figure 3b). These data were also provided by the Lithuanian company UAB"Inžinerinis projektavimas". The design enables precise reconstruction of the bridge piers and abutments in the physical as well as numerical models.

The hydrologic data were provided by the UAB"Inžinerinis projektavimas", similar to previous geometric data. The Lithuanian Hydrometeorological Service elaborated the flows for testing according to the obligatory rules in Lithuania [46]. The calculations were performed based on observations made in the period 1992–2016 in the gauge station called Nemunas-Kaunas. This station is located below the modeled bridge (Figure 4a). The values for tests are presented in Table 1. These are two minimum flows and three maximum flows. The minimum flows include the discharges with the probability of exceedance of 95%. Two values are provided, where the first refers to normal conditions, and the second is related to the winter conditions with ice flow. The maximum flows include discharges with assigned probabilities of exceedance of 50%, 5%, and 1%. These are the floods with average return times of 2, 20, and 100 years, respectively.

The data provided by the UAB "Inžinerinis projektavimas" also includes the samples of bottom sediment taken along the reach from the dam of the Kaunas reservoir to the village called Vilkija (Figure 1b,c). Near the bridge, two sampling points are located. In Figure 4a, these are points denoted as S-12 and S-13. Unfortunately, the long-term erosion processes below the Kaunas dam induced strong armoring of the river bottom below. The examples of the material found at the bottom are presented in Figure 4b. These are stones of different sizes up to 10 cm. Such a bottom should not be treated as alluvial. It is rather a stable bottom, but the assessment of the bed roughness is not possible based on sediment samples. Hence, this factor should be treated as uncertain.

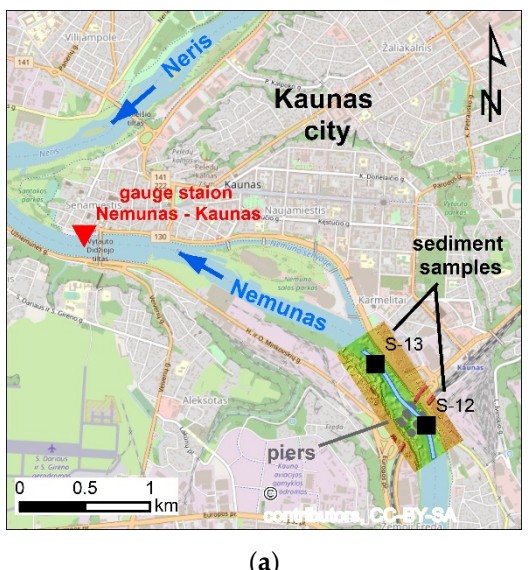
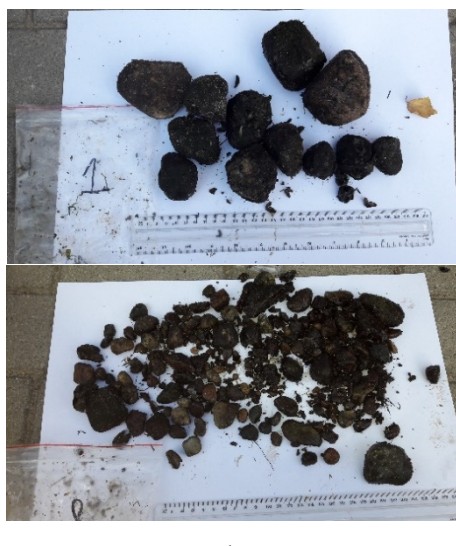

| (**a**) | (**b**) |

**Figure 4.** Gauge station and sediment samples: (**a**) location of gauge station and sampling points around the bridge, (**b**) examples of the material taken from the bottom.

**Table 1.** Provided hydrologic information.

| Type | Minimum Flows | | Maximum Flows | | |
|---|---|---|---|---|---|
| probability of exceedance | 95% | | 50% | 5% | 1% |
| specific conditions | without ice | with ice | | - | |
| symbol | $Q_{95\%}$ | $Q_{95\%,ice}$ | $Q_{50\%}$ | $Q_{5\%}$ | $Q_{1\%}$ |
| value (m$^3$/s) | 71.6 | 91.5 | 1212 | 2143 | 3079 |

As it is presented in Table 1, the range of tested flows is rather wide. In the case of low flows, the requirements on the acceptable depth are the most important. In the analyzed reach, these conditions are presented in Table 2. According to the table, the reach from the Kaunas dam to Vilkija is split into two parts. The border between them is the islands in the city of Kaunas, visible in Figure 4a. In the upper part, the requirements for the available depth are less restrictive. The depth should not be smaller than 1.15 m. In the downstream part, the required depth is greater and equals 1.40 m. The reason for lower restrictions in the upper part may be the functioning of the bridges in the city of Kaunas. So, it seems to be reasonable to test the flow conditions below the bridge with two required depths.

**Table 2.** Required depth along the entire reach under investigation. (*source: State Enterprise Lithuanian Inland Waterways Authority*).

| Condition | From | To | Required Depth |
|---|---|---|---|
| (1) | the Kaunas dam | the island in the city of Kaunas | 1.15 m |
| (2) | the island in the city of Kaunas | the village of Vilkija | 1.40 m |

As mentioned above, the Froude numbers are also tested, though, it is not required by any EU or specific Lithuanian documents. However, the basic hydraulic experience suggests that the tested criteria should not only be related to the depth in the large flowing river. Some criteria referring to the flow velocity should also be specified. Hence, it was assumed that the Froude number defined below is the proper measure of safety:

$$Fr = \frac{|u|}{\sqrt{gh}} \tag{1}$$

where $|u|$ is magnitude of local flow velocity, $g = 9.81$ m/s$^2$ is gravity acceleration, and $h$ means local depth. It was assumed that the value of the Froude number less than 0.50 means rather safe flow conditions. A value between 0.50 to 0.75 indicates a medium risk. If the Froude number is greater than 0.75, the flow conditions are rather dangerous. A totally unacceptable situation appears when the Froude number is close to or above 1.0. It means critical and supercritical conditions of flow with high velocity and relatively small depth. According to the author's knowledge and experience, no navigation should be done when such conditions appear anywhere in the channel. The assumed qualitative criteria for the Froude number are summarized in Table 3.

**Table 3.** Classes of Froude # taken for the analysis of flow conditions.

| Froude # | | Risk |
| --- | --- | --- |
| min | max | |
| 0.00 | 0.50 | no risk detected |
| 0.50 | 0.75 | medium risk |
| 0.75 | - | high risk |

The additional data used to support work in the GIS environment include Open-StreetMap [47], which is visible in Figures 3b and 4a. The photos taken from Google Street View [48] are presented in Figure 2. For some preliminary analyses, Google Maps [49] was also used. A topographic map provided by the ESRI company and available in the ArcGIS software databases was used. It is shown in Figure 1a. The last important layer is the ortophotomap. It is visible in Figures 1b and 3a. This layer was provided by the UAB "Inžinerinis projektavimas".

It is important to note at the beginning that the rules of scaling were implemented for physical modeling [50–53]. The rules of geometric, kinematic, and dynamic similarities were satisfied in constructing the physical model and planning the experiments in the hydraulic laboratory. In the case of the bridge in Kaunas city, the first two mean that the relationships between the main dimensions and velocities have to be the same in nature and in the model. The third criterion requires the same in the case of the main forces. It led to the requirement of the same Froude number in the two objects, natural and laboratory. The distorted geometric scale was applied, which means that the horizontal and vertical scales differed. In Table 4, the horizontal scales of length and width are denoted as $\lambda l$ and $\lambda b$, respectively. Their value is the same and equals 200. The vertical scale $\lambda h$ is 50. The other scales presented in Table 4 were linked to the basic geometry and mentioned similarities and requirements of the same Froude number in the model and in nature. The relationships and values obtained are listed.

The physical model was created in the Water Laboratory of the Department of Hydraulic and Sanitary Engineering at Poznan University of Life Sciences, Poland. Before the models were created, the measurements of the bathymetry were used to reconstruct a new channel bed. The spatial interpolation methods available in the ArcGIS extension called Spatial Analyst were tested, e.g., [54]. Finally, the natural neighbor method was implemented. The results are visible in Figure 5a. The 3D Analyst extension was applied for the creation of 11 cross-sections. In Figure 5a, these cross-sections are denoted as p-1, p-2, etc. The distance between cross-sections is constant, and it equals 100 m. They properly represent the variability of the bathymetry and terrain in the investigated area. The preparation of the model geometry is illustrated in Figure 5b–d. The sizes of the modeled area are 450 m × 1000 m. Applying the horizontal scale factor, the required size of the physical model is 2.25 m × 5 m. The sizes of the box where the model was located are greater and

equal to 3.25 m × 10 m. It enables the location of the inflow and outflow facilities, as well as the measurement devices. The cross-sections defined in ArcGIS, scaled according to the previous assumptions, were cut from plastic and located at the proper distances in the box (Figure 5b). The space between the cross-sections was filled with concrete. The surface was smoothed, then covered with a layer of glue and painted (Figure 6c) to reconstruct roughness in nature reflecting the possible range of values. This was problematic due to the issue mentioned earlier. Finally, the piers, abutments, and spur dikes were formed and put in proper locations (Figure 5d).

**Table 4.** Factors applied to scale the model.

| Type of Scale | Symbol | | Relationship with Other Factors | Value [-] |
|---|---|---|---|---|
| | Basic | Scaling Factor | | |
| length | $L$ | $\lambda_l$ | - | 200 |
| width | $B$ | $\lambda_b$ | - | 200 |
| height | $h$ | $\lambda_h$ | - | 50 |
| velocity | $u$ | $\lambda_u$ | $\lambda_h^{1/2}$ | 7.071 |
| discharge | $Q$ | $\lambda_Q$ | $\lambda_h^{3/2}\cdot\lambda_b$ | 70,711 |
| time | $t$ | $\lambda_t$ | $\lambda_l\cdot\lambda_h^{-1/2}$ | 28.3 |
| velocity of settling | $w$ | $\lambda_w$ | $\lambda_h^{3/2}\cdot\lambda_l^{-1}$ | 1.77 |
| pressure | $p$ | $\lambda_p$ | $\lambda_h$ | 50 |
| roughness size | $k$ | $\lambda_k$ | $\lambda_h^4\cdot\lambda_b^{-3}$ | 0.781 |
| Manning's roughness | $n$ | $\lambda_n$ | $\lambda_h^{2/3}\cdot\lambda_l^{-1/2}$ | 0.9597 |

It is assumed that the friction of the physical model surface is very close to the friction of the rough concrete. The suggested values should be about 0.011–0.015 s·m$^{-1/3}$. These values in the model reflect the roughness coefficients in nature at the level 0.012–0.016 s·m$^{-1/3}$. Such values may reflect the conditions of the armored bottom. However, applying the numerical model enables the verification of these assumptions.

The important element of the experiment's planning is the proper recalculation of inflows. The values from Table 1 and discharge scale factors from Table 4 are used. The original and recalculated values are presented in Table 5. Finally, the physical experiments were conducted with discharges $Q_{95\%}$, $Q_{5\%}$, and $Q_{1\%}$.

**Table 5.** Recalculation of discharges for the physical model.

| Type | Symbol | Unit | $Q_{95\%}$ | $Q_{5\%}$ | $Q_{1\%}$ |
|---|---|---|---|---|---|
| nature | $Q_N$ | m$^3$/s | 71.6 | 2143 | 3079 |
| model | $Q_M$ | dm$^3$/s | 1.013 | 30.306 | 43.543 |

During the experiments in the laboratory, the measurements of the water surface in cross-sections indicated in Figure 5a were performed. The average velocity in the cross-sections was assessed based on discharge measurements and the determination of the cross-section flow area. The last element was determined based on the model geometry. In experiments with greater discharges, $Q_{5\%}$, and $Q_{1\%}$, the additional water head was imposed in the outlet of the model to make the measurements easier. The water surface measurements and flow velocity estimations were recalculated into actual conditions in the natural object. Such values are applied to validate the model before the major computations.

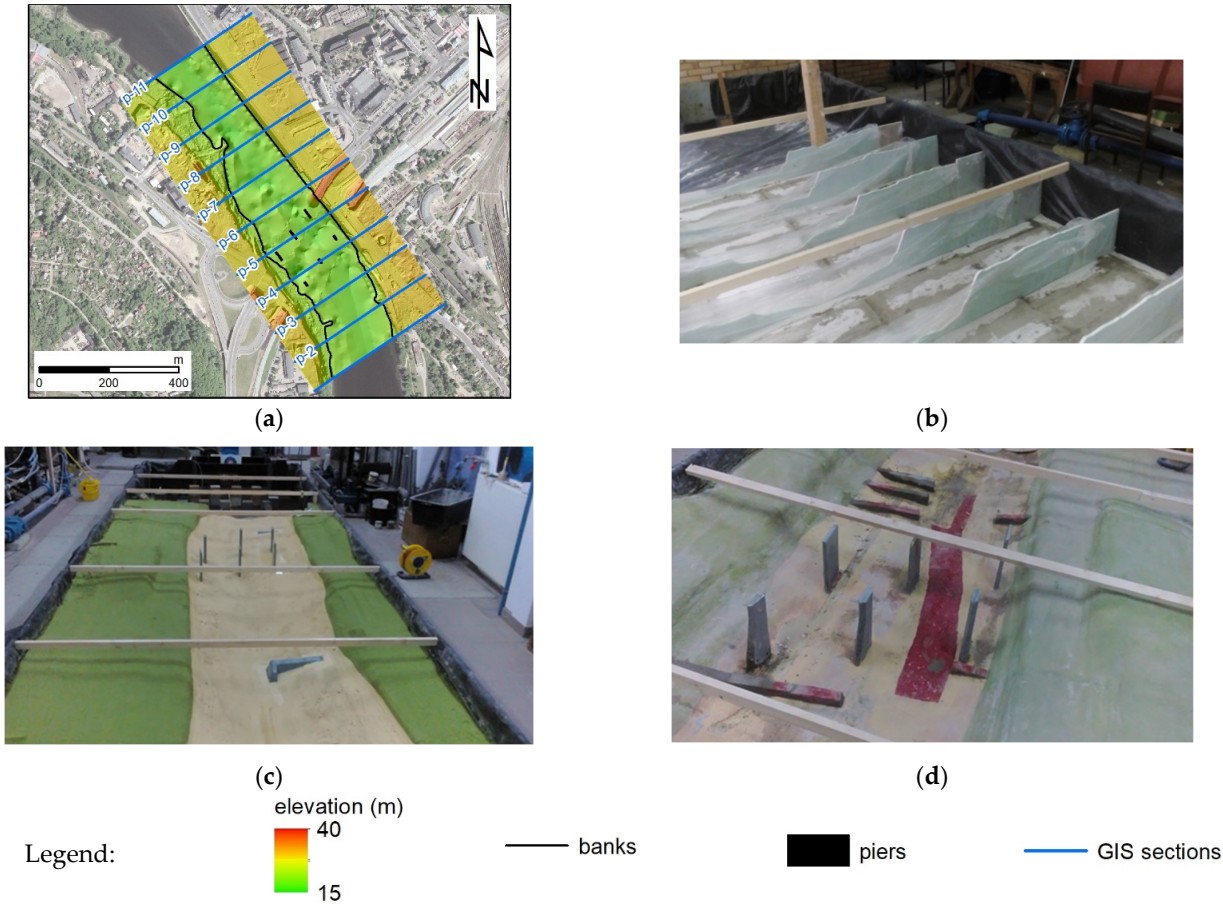

**Figure 5.** Physical model preparation: (**a**) DTM with interpolated bathymetry, (**b**) the cross-sections, (**c**) the final version, (**d**) piers, spur dikes, and waterway.

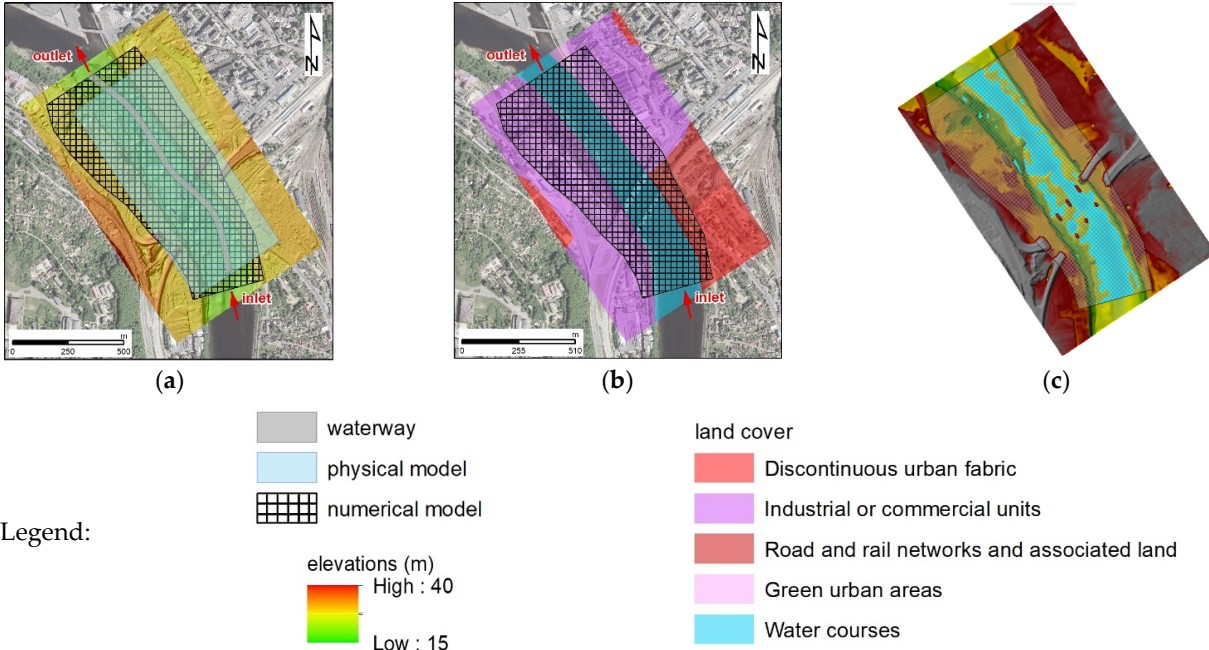

**Figure 6.** Geometrical properties of the numerical model: (**a**) the modeled area with additional elements, (**b**) the land cover from Corine 2018 database, (**c**) the modeled area in HEC-RAS.

The numerical model was created with the help of the HEC-RAS software in version 6.3.1 [55–57]. The ArcGIS 10.8.2 with plug-in HEC-GeoRAS [58] was used to prepare the geometry of the computational model. The model only consists of a single 2D area with a single inflow and outflow (Figure 6a). The HEC-GeoRAS with DTM was used to define the standard Storage Area in ArcGIS. The region for numerical modeling is not the same as the area covered by the physical model. The preliminary results of the experiments were used to extend the numerical model where necessary and shrink the area where it is possible to avoid too high a computational time. After exporting to HEC-RAS, it was converted to a 2D area with over 126,000 cells of the base size 2 m × 2 m (Figure 6a–c). Due to the mechanism of mesh generation, the cells near the boundaries have shapes different than squares, and their sizes could differ from the base cells. The physical model has a uniform roughness close to concrete, as it was mentioned before. Hence, the initial land surface cover was also treated as uniform and implemented in the model applied in the validation process. In the next computations, the roughness is distributed according to the land covers loaded from the Corine 2018 database for Europe [59] (Figure 6b). In the modeled area, there are five different types of covers. These are listed in Table 6. For the base covers denoted in the Corine database as 112, 121, 122, and 141, the choice of the Manning's coefficient is not so crucial. The assumed values are also presented in Table 6. The most important is the roughness, denoted as 511 and representing the riverbed.

**Table 6.** Initial values of roughness coefficients assigned to land covers according to the Corine 2018 database.

| Corine Code | Type of Land Cover | Roughness Coefficient (s m$^{-1/3}$) |
| :---: | :---: | :---: |
| 112 | Discontinuous urban fabric | 0.035 |
| 121 | Industrial or commercial units | 0.035 |
| 122 | Road and rail networks and associated land | 0.020 |
| 141 | Green urban areas | 0.045 |
| 511 | Watercourses (river bottom) | 0.012–0.015 * <br> 0.015–0.060 ** |

Notes: * the range of bed roughness tested in the process of model validation; ** the range of bed roughness tested in the major computations of the waterway safety.

Due to the mentioned armoring of the bed, the preliminary roughness applied for this zone was in the range of 0.012–0.015 s·m$^{-1/3}$. This value is very small and fits the concrete surfaces like the one applied in the physical model. The measurements converted from the physical model are compared with computations to assess if the numerical model is suitable to properly reconstruct flow phenomena. The assessment was performed with an RMSE—relative mean square error.

$$RMSE_\varphi = \sqrt{\frac{1}{N} \sum_{i=1}^{N} \left[ \frac{\varphi_i^{(s)} - \varphi_i^{(m)}}{\Phi_i^{(m)}} \right]^2} \times 100\% \tag{2}$$

where $N$ is a number of compared values, and $\varphi_i$ are the values, e.g., water level or average velocity. $\Phi_i$ is the reference value applied to express the differences in percentages. In the first case, when water surfaces were compared, the maximum depth in the cross-sections was considered. In the case of the velocity, the values estimated for the measurements are used. The superscripts (s) and (m) mean simulation and measurements, respectively. It is important to mention that the averaged velocities for the measurement were estimated on the basic of discharges and calculation of flow area from the water levels and topography.

The averaged velocities from the calculations were estimated on the basis of velocity magnitudes read along the cross-sections. The latter is quite simple and fits the formats of results easily exported from HEC-RAS. However, it may produce some errors when the flow is not longitudinal, but more variable in space. Due to that, the verification of the velocity range is done only for cross-sections where the flow is expected to be 1D. These are p-1, p-2, and from p-7 to p-11. The cross-sections of the bridge (p-4, p-5), as well as one cross-section before (p-3) and one after (p-6), were excluded.

Next, the values 0.015, 0.025, 0.035, 0.050, and 0.060 s·m$^{-1/3}$ were tested. These conditions represent conditions of greater friction, which may be caused by irregularities in the river bed. In general, steady flow computations are required if the purpose of the analysis is considered. The majority of the available 2D river flow models enable computations in unsteady flow conditions, which may be easily applied to reconstruct steady flow. In this case, the same approach was used. HEC-RAS gives access to different models describing the overland flow. The simplest, fastest, and most stable is the diffusive wave model. The equations applied there do not include the inertia terms. Such an approach may be correct for many applications, but in some cases, it may provide inaccurate results. Hence, this model was used only to generate the initial condition representing preliminary steady flow. The second model is based on the Shallow Water Equations and includes all necessary terms, namely inertia, pressure, gravity, and friction. The second model starts with the initial condition generated as the final state of the computations with the previous model. Such an approach accelerated the computations and guaranteed the required accuracy.

The boundary conditions applied in each model include inflow and outflow. As the inflow, the hydrograph with the constant flow is applied. These flows are presented in Table 1. In these simulations, the free flow condition is used in the outlet to determine the flow conditions in the waterway. This condition is imposed as the Manning equation with a specified slope equals 0.3‰. This is the estimated value of the bottom slope in the outlet of the model. The results generated are raster layers of depth and velocity components exported to GeoTIFF format and then processed in ArcGIS. On this basis, the requirements of the depth and threat related to the velocities is analyzed along the waterway course in the modeled areas.

The general scheme of analyses conducted, including linking simulations and laboratory tests, is presented in Figure 7. The numerical and physical models consist of two parallel branches developed based on the same data. The first experiments influence the choice of the numerical modeling area. Then, the results of the laboratory experiments are used to validate the results of the simulations. Such a procedure was elaborated because it was assumed that the direct measurements of the water level in the analyzed object are not available for calibration of any model, neither physical nor numerical. In general, this approach is compatible with the fact that the analysis is related to extreme conditions, minimum as well as maximum flows. In such cases, any application of the model would be outside the ranges of calibration due for obvious reasons. On the other hand, the approach presented is good enough in cases when the object does not exist, e.g., in the design phase. In the next step, the numerical model enables the extension of the analyses and testing of the different roughness in the bottom. The depth maps and maps of the Froude number distributions are generated from the results of the hydrodynamic computations. And finally, the maps are reclassified according to the criteria defining the levels of risk.

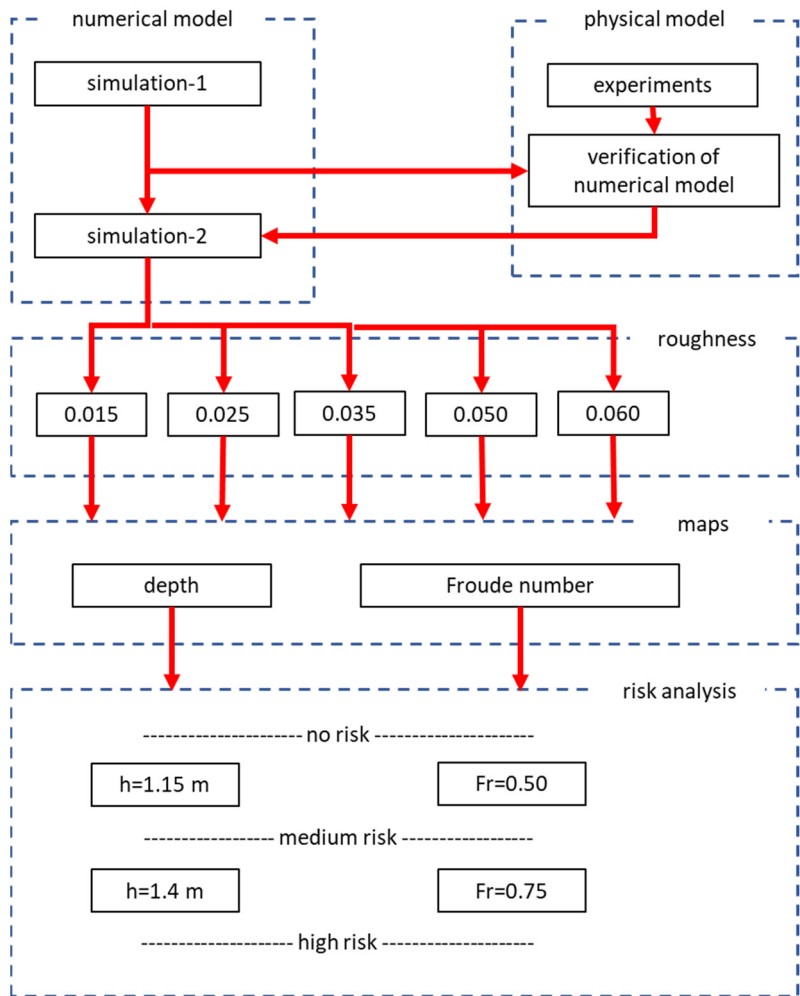

**Figure 7.** Block scheme of the designed analysis steps linking hydrodynamic simulations, laboratory tests, and geoprocessing.

## 4. Results

A few runs with different roughness coefficients were tested in the validation phase. The value providing the satisfactory fitness of the measurements and the computations is 0.012 s·m$^{-1/3}$. An example of the obtained results is shown in Figure 8. These are the results for simulation with $Q_{5\%}$. On the left (Figure 8a), the comparison of measured and computed water surface elevation can be noted. Above the red dots representing the measurements, the numbers of the cross-sections are denoted. The numbers are consistent with Figure 5a. The letter R added to cross-sections 4 and 5 means the right side of the cross-sections, which are split by the piers. The compatibility of water surfaces seems to be good. It is also confirmed in the values of RMSE presented in Table 7. The relative error for the water level expressed in % varies between 1.04 and 3.33%.

In Figure 8b, the averaged velocities are presented. The cross-section denotations are shown as the labels of the vertical axis. In this case, the results are quite good in some cross-sections and a little bit worse in others. Satisfactory compatibility was obtained for these cross-sections, where the flow was more or less one-dimensional. These are listed above, where the definition of the RMSE is explained. For these cross-sections, the results obtained are also shown in Table 7. The errors of the average velocity vary between 16.92 and 21.34%. The values obtained are not satisfactory, but the comparison of assessed and simulated average velocities in Figure 8b explains the reasons. The worse results are achieved in the cross-sections p-7, p-8, and p-9. A view of these cross-sections in Figure 5a suggests that the flow there may not be 1D as it was assumed a priori. Considering all circumstances, the

results are considered satisfactory. The numerical model seems to be good enough to make the next tests.

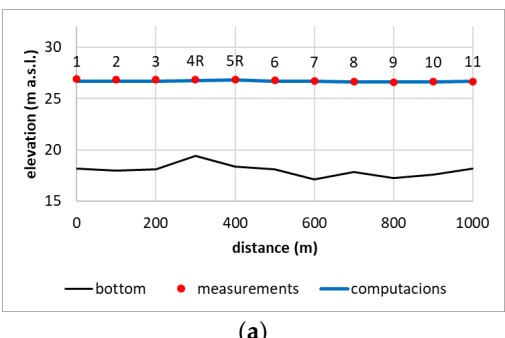 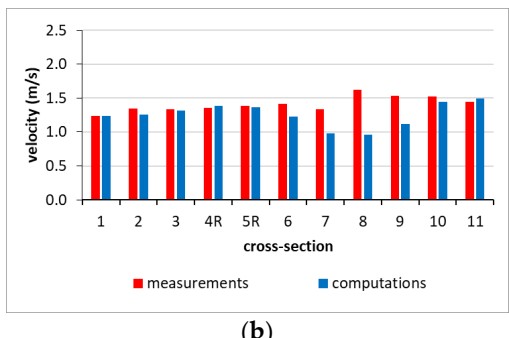

(**a**)                    (**b**)

**Figure 8.** Graphical illustration of the model validation example: (**a**) water surface profile for $Q_{5\%}$, (**b**) estimations of mean velocities for $Q_{5\%}$.

**Table 7.** Final results of model validation calculated for cross-sections where the flow was one-dimensional (p-1, p-2, and from p-7 to p-11, as shown in Figure 5a).

| Type of Experiment | RMSE (%) | |
| :---: | :---: | :---: |
| | **Water Surface Elevation** | **Mean Velocity** |
| $Q_{95\%}$ | 3.33 | 16.92 |
| $Q_{5\%}$ | 1.22 | 21.34 |
| $Q_{1\%}$ | 1.04 | 20.34 |

The main results are distributions of depths and maps of Froude numbers in the modeled area with special emphasis on the course of the waterway. The analysis of these two factors is presented in the form of maps (Figures 9–11), profiles (Figure 12), and summary graphs (Figure 13). The most crucial results are minimum depths and maximum Froude number. The specific values are presented. Additionally, the distance with violation of basic criteria discussed in the previous section is also shown.

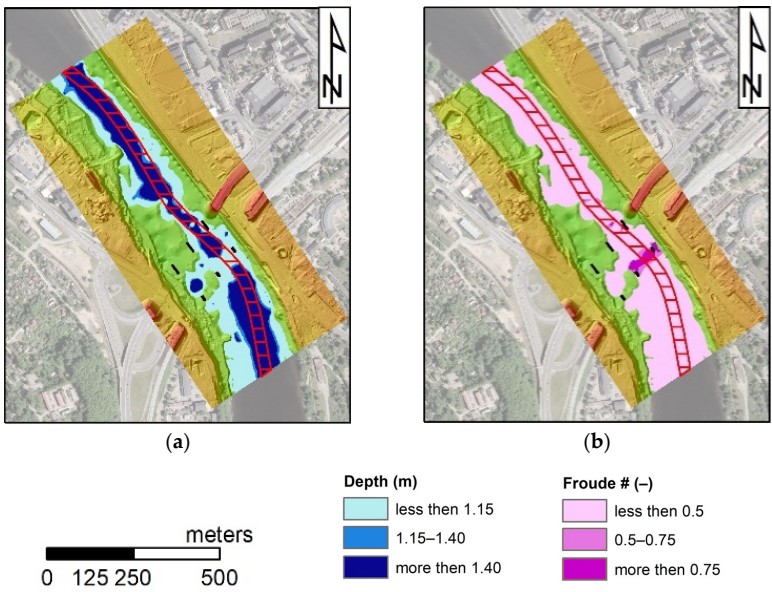

(**a**)                    (**b**)

**Figure 9.** Example of results obtained for the roughness of the bottom equal to 0.025 s m$^{-1/3}$ and flow $Q_{95\%}$. Maps of the entire model with depth and Froude #: (**a**) map of depth classes, (**b**) distribution of Froude #.

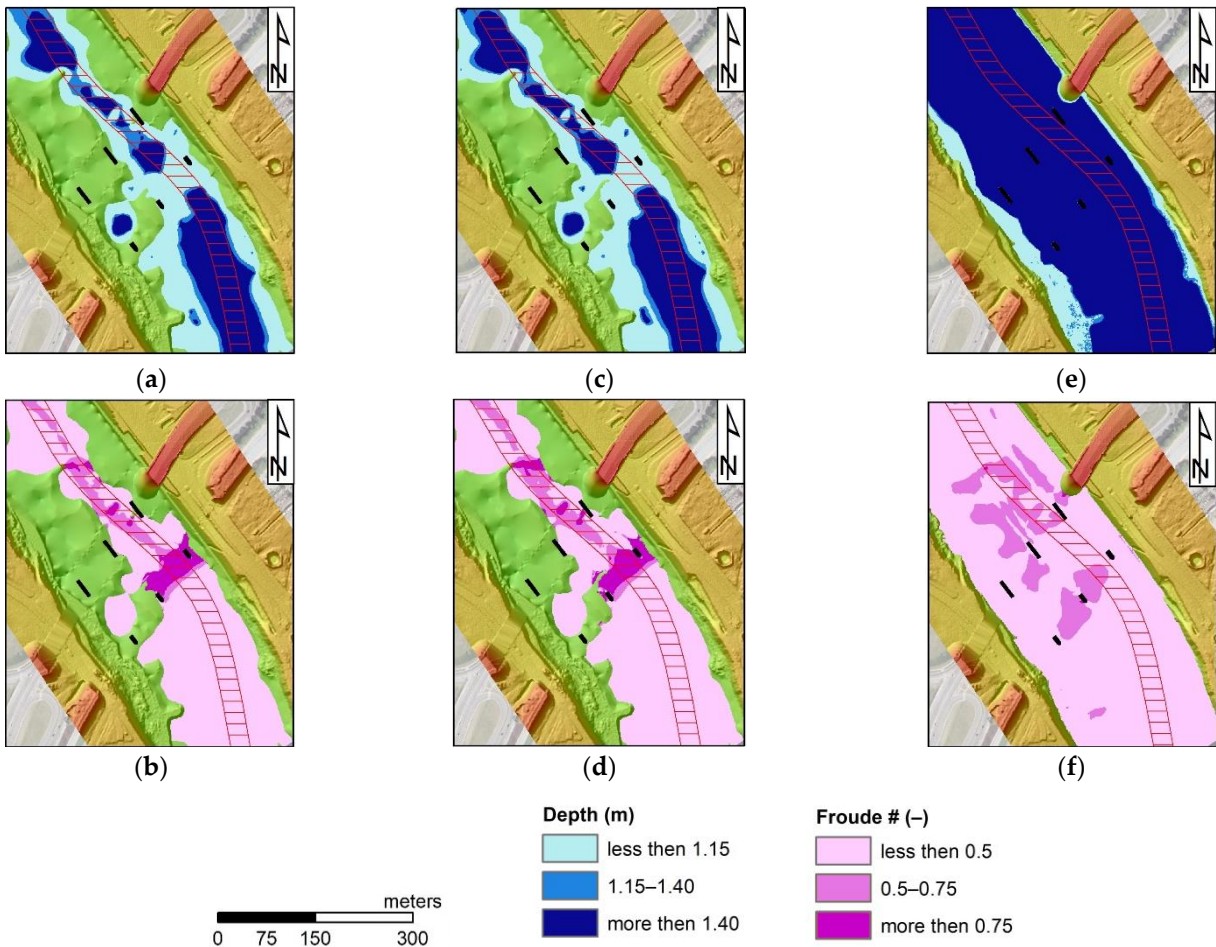

**Depth (m)**

less then 1.15

1.15–1.40

more then 1.40

**Froude # (–)**

less then 0.5

0.5–0.75

more then 0.75

meters
0    75    150          300

**Figure 10.** Results obtained for the roughness of the bottom equal to 0.015 s m$^{-1/3}$: (**a**) map of depth classes for $Q_{95\%}$, (**b**) distribution of Froude # for $Q_{95\%}$, (**c**) map of depth classes for $Q_{95\%,ice}$, (**d**) distribution of Froude # for $Q_{95\%,ice}$, (**e**) map of depth classes for $Q_{50\%}$, (**f**) distribution of Froude # for $Q_{50\%}$.

The first example is presented in Figure 9. These are results obtained in the simulation of $Q_{95\%}$ flow with roughness 0.025 s·m$^{-1/3}$. In Figure 9a, the map of depths in the entire area of the physical modeling was presented. On the right side, the distribution of the Froude number in the same region is illustrated. The waterway is denoted as a belt of red lines. It is well visible that the most severe conditions are detected in the inlet to the bridge section, between the piers close to the right bank; i€ this case, a depth less than 1.15 m and Froude number greater than 0.75. However, there are also other locations where the assumed criteria are violated.

A closer view of the most dangerous area is presented in Figures 10 and 11. These are maps of depth and Froude numbers obtained for three tested flows, $Q_{95\%}$, $Q_{95\%,ice}$, and $Q_{50\%}$. The results presented in Figure 10 were obtained in the simulation with the minimum value of bed roughness applied in numerical tests. This is 0.015 s·m$^{-1/3}$. The next set of results represents conditions with maximum assumed roughness of the channel bottom, which equals 0.060 s·m$^{-1/3}$. The visible differences between these two simulations indicate the stabilizing influence of the higher friction.

The results visible in the first figure (Figure 10) show well the range of the most dangerous zone below the bridge. The depth is too small in the computations with two low discharges, $Q_{95\%}$ (Figure 10a) and $Q_{95\%,ice}$ (Figure 10c). In the case of $Q_{50\%}$, the amount of flowing water prevents this risk (Figure 10e). The areas of low depth are the same as those of too-high Froude numbers in Figure 10b,d. It is well shown that the low depth favors greater velocity, which finally results in high values of the Froude number. In the case of

the flow $Q_{50\%}$, the proper conditions for the depth do not mean total safety. In Figure 10f, medium risk related to the values of the Froude number between 0.50 and 0.75 was shown.

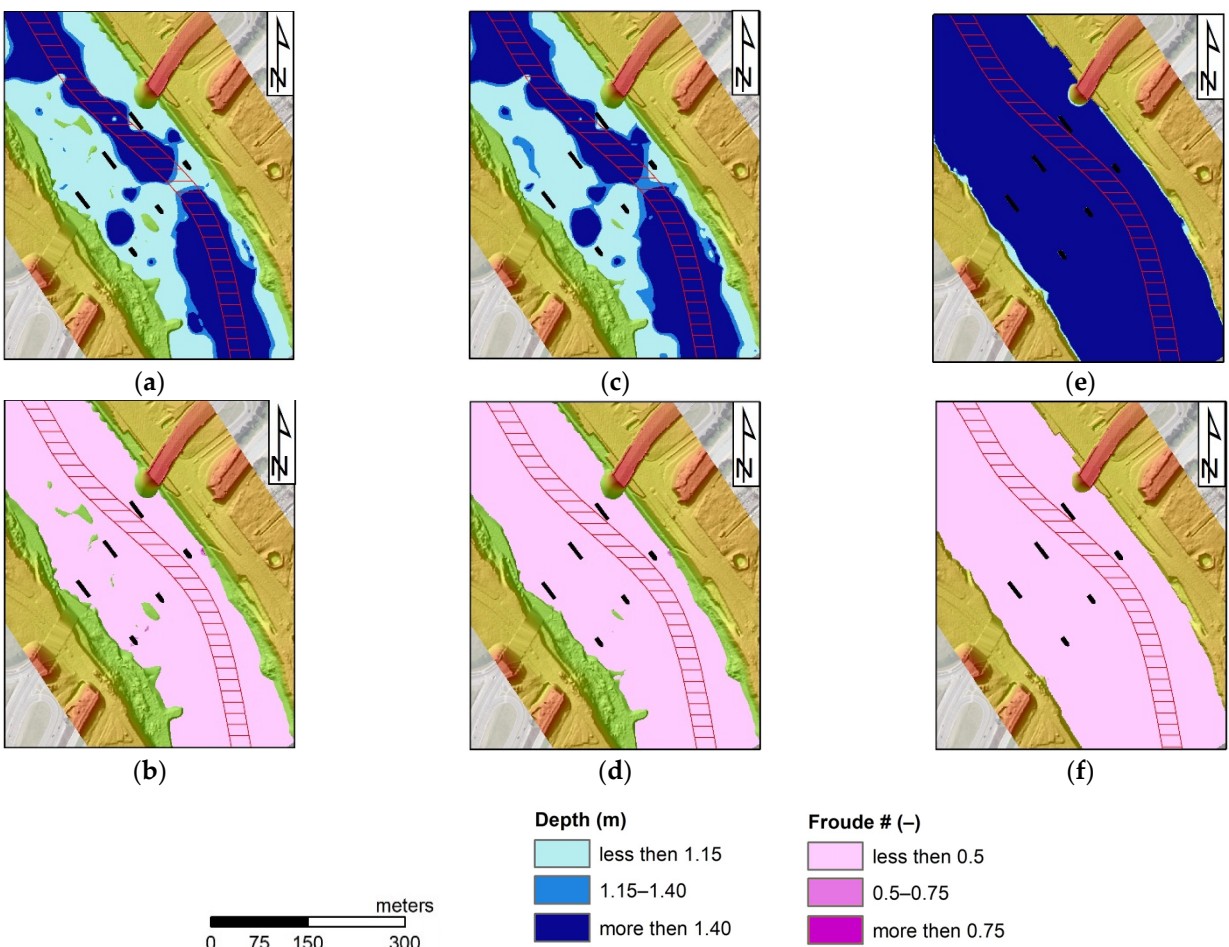

**Figure 11.** Results obtained for roughness of the bottom equal to 0.060 s m$^{-1/3}$: (**a**) map of depth classes for $Q_{95\%}$, (**b**) distribution of Froude # for $Q_{95\%}$, (**c**) map of depth classes for $Q_{95\%}$, ice, (**d**) distribution of Froude # for $Q_{95\%}$, ice, (**e**) map of depth classes for $Q_{50\%}$, (**f**) distribution of Froude # for $Q_{50\%}$.

These distributions of values under consideration and patterns of safe and dangerous zones are changing with the roughness changes in the numerical model's computations. A good example is Figure 11. In the case of the maximum tested roughness, the location of the zones with too low depth is similar, but the sizes of these areas are significantly lower (Figure 11a,c). Evidently, greater friction and energy losses increase the water stages. In Figure 11b,d, it is shown that these factors also affect the velocities and Froude numbers. The risk of this type is not detected in this case.

To illustrate better the character of the danger and risk below the bridge, the profiles of the water surface and Froude numbers were prepared. The example is shown in Figure 12. These are results obtained in the computations with discharge $Q_{95\%,ice}$. The first graph (Figure 12a) includes the water surface profiles with the profile of the bottom. The second graph (Figure 12b) shows the changes of the Froude number along this part of the waterway. In both figures, the bridge piers are denoted as gray areas. The boundaries of the depths described in Table 2 and assumed levels of the risk related to the Froude number explained in Table 3 are also marked in Figure 12. It is well visible that the depth requirements are violated between the first row of piers. The main reason is the increase in the bottom elevations in this zone. It could be caused by the past accumulation of the sediments transported as the effect of erosion below the Kaunas dam. A similar violation

is also observed below the bridge. There are also higher bottom elevations. In this case, the reason may be the accumulation of sediments generated by erosion between the two rows of the piers. It is also visible that between the mentioned rows of piers, the bottom elevations are significantly lower. Figure 12b with Froude number profiles also indicates the dangerous conditions in the same sections, between the first row of piers and below the bridge. In the second case, the high Froude number suggests that there are conditions prone to the generation of the hydraulic jump. However, such danger should be detected with more detailed analyses, including 3D flow simulations. Despite this, the occurrence of the conditions computed for $Q_{95\%,ice}$ in reality, should stop all water transport along this reach of the river.

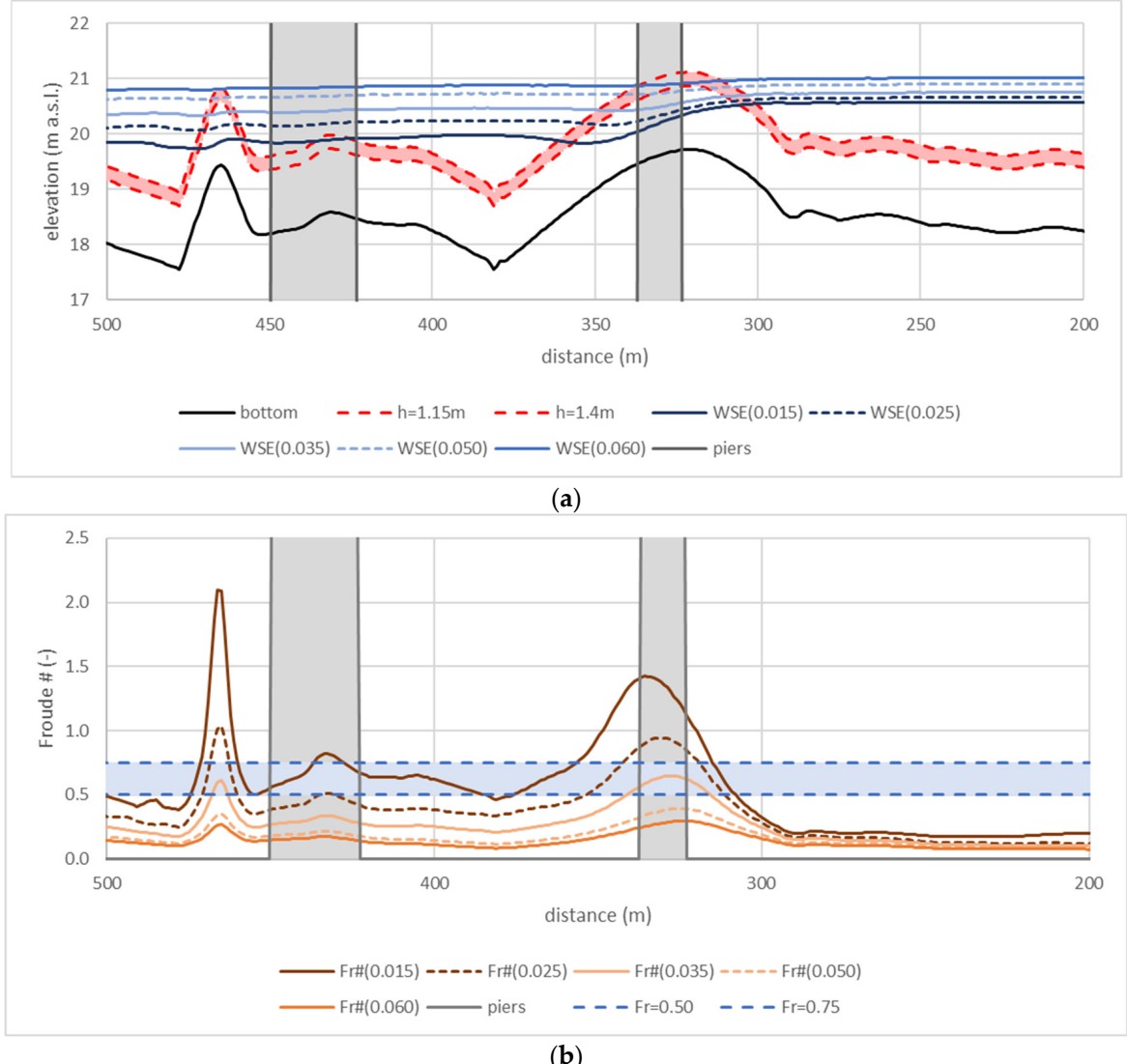

**Figure 12.** Examples of profiles under the bridge for discharge Q95%, ice: (**a**) water surface, (**b**) distribution of Froude #.

Summary results are presented in four graphs in Figure 13. The first two, in Figure 13a,b, are related to the depth. The next present the Froude number or connected measures. All the results were produced based on the computations with low discharges $Q_{95\%}$ and $Q_{95\%,ice}$. The results obtained in tests with $Q_{50\%}$ were not indicating important hazards along the analyzed reach. Hence, they are not applied in the final discussion.

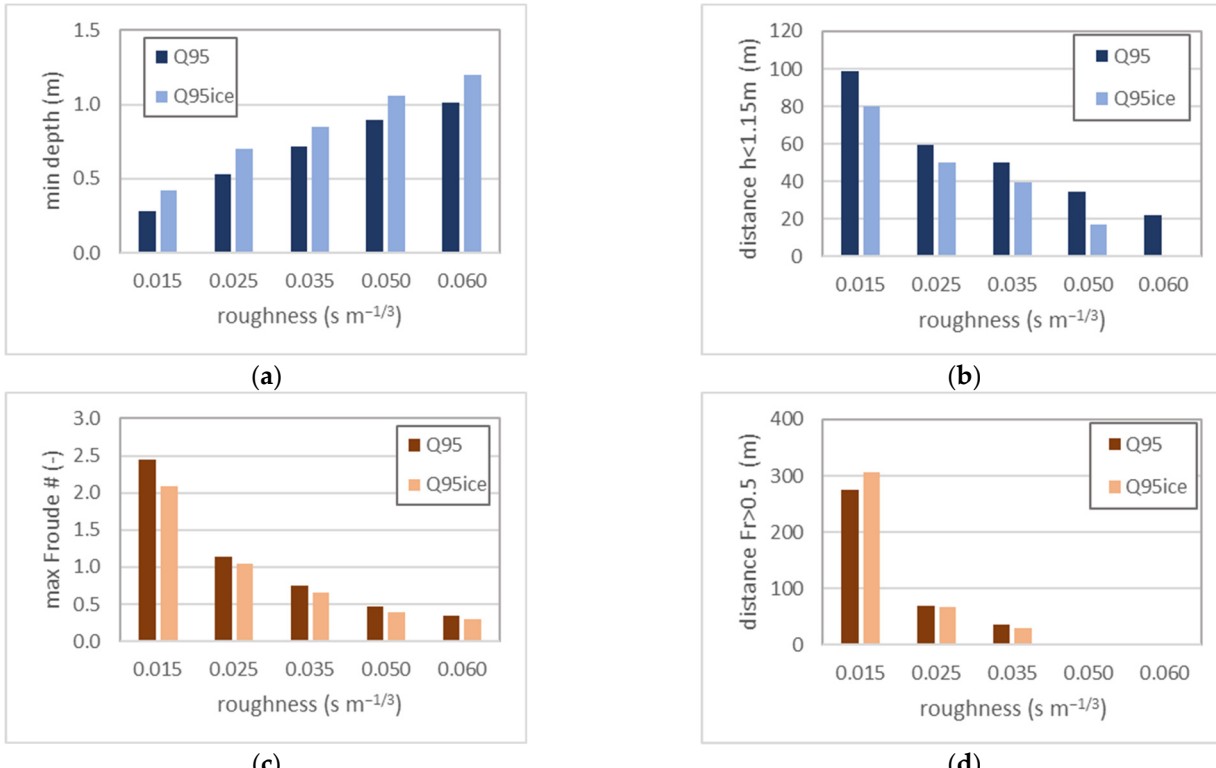

(**a**)  (**b**)

(**c**)  (**d**)

**Figure 13.** Summary of results for $Q_{95\%}$ and $Q_{95\%,ice}$: (**a**) minimum depth, (**b**) the distance with the depth below 1.15 m, (**c**) maximum Froude #, (**d**) the distance with Froude # above 0.5.

In Figure 13a, the minimum depths are presented. These results were obtained during the computations with different roughness coefficients selected for the riverbed. Obviously, the minimum depth for $Q_{95\%,ice}$ is slightly greater than the same indicator calculated with $Q_{95\%}$. The difference in the magnitudes of the discharges is a proper explanation of this fact. Additionally, it may be noted that the minimum depth increases with the increasing roughness. It confirms the previous remark on the stabilizing function of friction. It could be treated as a potential solution to the problems related to the functioning of the waterway, but it should be remembered that there is another side of this phenomenon. The greater energy losses caused by higher roughness are also responsible for the increase in the water surface elevations during a flood and wider inundation areas. The consequences may be more expensive flood losses. Hence, the increase in the roughness should not be recommended as the method for coping with two small depths in existing waterways.

Figure 13b shows the results related to the depth distribution along the investigated channel. These are the distances with a depth less than the required 1.15 m (Table 2). This distance for the lowest roughness 0.015 s·m$^{-1/3}$ equals 98.83 m, which is approximately 10% of the waterway reach modeled in this research. This relatively large ratio decreases with the increasing energy losses related to the greater friction. In the case of the greatest roughness coefficient of 0.060 s·m$^{-1/3}$, such adverse conditions are observed along 21.72 m if the lowest flow $Q_{95\%}$ is analyzed. Hence, this distance decreases about five times with an increase in the roughness of four times. In the computations with the $Q_{95\%,ice}$, depths lower than 1.15 m were not observed.

The analysis of Froude numbers in the same computations provided a slightly different result. As may be seen in Figure 13c, the lowest roughness values, both 0.015 s·m$^{-1/3}$, and 0.025 s·m$^{-1/3}$, serve well potential supercritical conditions. This is very dangerous even if the depths are satisfactory. The flow conditions are relatively risky, with the Froude number above 0.50 appearing along more than 300 m in the test with $Q_{95\%,ice}$, which is about 30% of the analyzed reach (Figure 13d). Fortunately, this problem vanishes quickly with the increase in the roughness, which is also well illustrated in Figure 13d.

## 5. Discussion

Analyzing hydraulic structure behavior with physical modeling is a common approach in hydraulic engineering. An excellent example of a similar approach was presented by Szydłowski [27]. The author analyzed flow under the hypothetical bridge with simplified piers. The main investigated elements were distributions of depth, velocity, and turbulent structures. Because the basis for this work was not a real bridge existing in nature, the author focused only on the impact of the piers on the flow structure. In our work, coupled interactions between the bridge construction and irregular bed are also included. However, the common element of research made by Szydłowski [27] and the presented work is the opportunity to generate very dangerous supercritical flow conditions below the bridge.

Physical modeling effectively applied here is also widely used in designing new waterways or planning necessary modifications to existing waterways. Such examples were presented in Menéndez et al. [16] (2014) and Ametller [60]. In both cases, the new elements of the Panama channel are tested on the laboratory scale. These elements are prepared for the control of hydraulic conditions in the channel. Direct control is not possible in the case presented here, but the physical model may be used for many purposes. The primary aim is to detect the hazards related to insufficient hydraulic conditions. The second obvious approach may be aimed at testing the necessary modifications to the waterway.

The numerical modeling enables the extension of the analysis scope and more precise identification of the threats. The concept of validating or verifying a numerical model based on physical modeling was also applied before in the analyses of waterway functioning. Muste et al. [33] present an effective combination of this type. Although the simplified kinematic flow model is applied in the mentioned work, the numerical experiments are an extension of the laboratory results. A similar approach may be seen here. Due to the development of numerical simulations, it was possible to implement a more sophisticated and accurate model. However, in both cases, the combination of physical and numerical modeling enables overcoming physical modeling constraints related to the availability of resources and time.

Modern numerical modeling is also a good tool for directly analyzing all roughness uncertainties in hydraulic systems. An example of such analysis was presented by Andersson et al. [61]. The authors applied sophisticated CFD models for the analysis of roughness impacts on flow structures in hydropower tunnels. The case presented here is different because the free surface flow is implemented. Hence, the HEC-RAS model based on the shallow water equations is better suited for the problem investigated. The numerical simulations enabled the analysis of different roughness values and observations of their potential impact on the flow in the investigated channel reach.

In general, CFD is also used in analyses directly related to the functioning of waterways. Due to the complexity of the full 3D models based on the CFD concepts, these are mainly implemented to analyze local effects related to turbulence structures or interactions between streams and ships. Examples were given by Kang et al. [62] and Kang and Sotiropoulos [63], who focused on turbulence effects in natural navigation channels. Later Huang et al. [18] applied a 3D model based on the numerical solution of Navier-Stokes equations to analyze the erosion of the channel banks caused by the ship's movement. Both approaches provide detailed maps of velocities, pressures, and other dependent variables. For the analysis of waterway safety, such a detailed analysis may not be necessary. The second very common approach in this field is the application of 2D models of different types. The combination of 2D depth-averaged and 2D vertical models was implemented by Guerrero et al. [64] for a waterway in the Parana River. The obtained results include the horizontal and vertical distribution of velocities. The analyzed reach of the Parana River is a rather shorter navigable channel. However, the computational effort made could be time-consuming. Another sophisticated concept is the application of a semi-3D model for the simulation of the hydraulic conditions in the Chicago waterway by Zhu et al. [65]. This reach is also short. Hence, applying such time-consuming and computer-loading

computations was successful, though simplification like hydrostatic pressure distribution seemed necessary.

In general, the channel reaches of navigable channels are modeled with 1D or 2D models based on shallow water models with depth-averaged equations. The example of the 1D approach implemented on a quite long reach was tested by Diwedar et al. [66]. Sometimes, the hydrologic models are used to analyze the functioning of the larger systems of the navigable channels like was presented by Scheepers et al. [67]. The 2D hydrodynamic approaches may be suited well to model shorter channel reaches, as shown in Suner and Bas [17]. Nevertheless, more obvious is an implementation of a 2D concept for the analysis of single structures like a separated bridge by Szydłowski [27] or the channel through Vistula Spit [68].

The approach presented here is based on the application of the 2D depth-averaged model for the analysis of flow conditions under a single structure and its surroundings. The results obtained include depth maps similar to Szydłowski [17] and Szydłowski and Kolerski [68]. However, the maps of velocities also applied in the mentioned works are not used in the research presented here. Instead, the maps of Froude numbers were elaborated, taking into account the specifics of the waterway and expectations related to assessing its safety. The Froude number represents more complex characteristics of the flow conditions, including the insight of velocity, possible turbulence, and opportunity to generate effects like hydraulic jump. Further, the results obtained from the hydrodynamic simulation were classified into conditions acceptable, risky, and non-acceptable. The combination of this approach with the GIS techniques enabled not only the identification of existing risk but also the precise determination of the location, though the roughness of the channel bed was treated as uncertain.

## 6. Conclusions

This research presents an advanced combination of methods for analyzing waterway safety under a bridge and the reach around this structure. The processing of GIS data is a preliminary step that involves the preparation of the physical and numerical model. The physical model was constructed on the basis of data carefully extracted from the DTM and bridge design documentation. The numerical model uses the same sources of information as the basis for the generation of the numerical mesh with elevations and proper roughness cover. The experiments with the physical model were conducted to verify the correctness of the numerical model. And finally, the handling of roughness in the numerical model enabled the extension of the analysis, which could be difficult with the physical model only. The analyses focused on the lack of knowledge about the values of the roughness coefficient. This is a typical problem in many hydraulic engineering studies.

In particular, the study showed that the analyzed bridge might cause a problem for the functioning of the waterway along this reach. For the low flow conditions, the risk that the depth is lower than required is high. Although this deficit and its spatial extent depend on the roughness, it was observed in all tested cases. Hence, it is difficult to imagine the higher degree of roughness that could cause the vanishing of this problem. Analysis of the Froude number proved that the risk related to the depth is linked with very dangerous velocity conditions. The problem with the too-high Froude number is noticed in the same locations where the too-low depth is observed. Additionally, the analysis of this factor indicates the risk of hydraulic jump formation. Such phenomena may occur below the bridge or between the piers of the upstream and downstream parts of the structure. This risk is also related to the roughness but vanishes faster than the hazard caused by too small depth.

The presented methodology may be easily applied in other cases, too. The combination of hydrodynamic simulations and geoprocessing in the stage of pre- and postprocessing could be a powerful tool in hydraulic engineering analyses. On the other hand, the simulations should be verified on the basis of real results. Sometimes it is difficult to conduct proper measurements. When the object of interest, e.g., a bridge or other structure, is only designed, it is simply impossible. In such cases, the support of physical modeling is

crucial. The linkage between numerical modeling and laboratory experiments is clearly presented here. Additionally, it is worth noting that numerical modeling enables a wider analysis of potential conditions than could be possible with a physical model only. Hence, the combination of methods of these two types gives real advantages.

**Author Contributions:** Conceptualization, T.D., T.K., K.M., J.W.-D. and J.V.; methodology, T.D., T.K., K.M. and S.K.; software, T.D.; validation, T.D., T.K., P.Z., S.Z. and N.W.; formal analysis, T.D., S.Z., N.W., S.K. and J.V. investigation, T.D. and P.Z.; resources, K.M., J.V., R.B., S.K., S.Z., J.W.-D., N.W., J.N. and T.K.; data curation, K.M., J.V., T.D. and S.Z.; writing—original draft preparation, T.D., T.K., J.W.-D., J.N., R.B., N.W. and P.Z.; writing—review and editing, T.D., J.W.-D., J.N., P.Z. and R.B. visualization, T.D. and J.W.-D.; supervision, T.D., T.K. and K.M.; project administration, T.K., T.D. and K.M.; funding acquisition, K.M. and J.V. All authors have read and agreed to the published version of the manuscript.

**Funding:** This research received no external funding.

**Data Availability Statement:** Data sharing is not applicable to this article.

**Conflicts of Interest:** The authors declare no conflict of interest.

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
