# Peer review of "Application of Physical and Numerical Modeling for Determination of Waterway Safety under the Bridge in Kaunas City, Lithuania"

_water, doi:10.3390/w15040731_

Round 1

Reviewer 1 Report

The manuscript can be accepted 

Reviewer 2 Report

Review of  Application of physical and numerical modeling for determination of waterway safety under the bridge in Kaunus City Lithuania

The following four deficiencies require correction before this paper is accepted for publication.

(1)   Absence of observed data on water surface profiles for the study reach

This paper presents results from physical and mathematical models of river flows in a reach of the Neumas River near Kaunus City Lithuania.  Both modeling procedures produce longitudinal water depth and water velocity profiles for various flowrates in the river.  In order for these model results to be accurate representations of how the river flow behaves it is necessary to calibrate each of the models to water surface profiles that have been observed in the river at known river flowrates.

The authors do nor present any data on water surface profiles that have been observed in the study reach.  Unless the authors can explain why the models have not been calibrated by comparison with observed water surface profiles and  how the model results they present are known to be correct representations of the behaviour of the river flows at this site without any comparison to observed river profiles the paper should not be accepted for publication.

To be more specific the river-flow depths in the reach under study for any specific flowrate depend on the water depth vs flowrate relationship at the downstream  end of the study reach. In the physical model and in the mathematical model the boundary condition of a known depth in the river at the downstream end of the reach for a specified flowrate must be present for the resulting modeled profile to be valid. In the physical model this is done by manipulating the tailgate waterlevel in the model to the known depth before measuring the depth profile. In the mathematical model the boundary condition at the downstream end of the reach is set as a start to calculations for the profile.

(2)   Additional Information is required on the specification of minimum depth

The required depths stated in Table 2 must be explained .    The source of the required depths is not cited – what organization has set these required depths ?    What is the draft of vessels using the two reaches ?  Also the explanation for the difference in required depth between the two sections of the reach is not satisfactory  - how does the “functioning of the bridges in he city” influence the minimum required depth ?

(3)   Operational experience of vessels using the reach for water transportation is missing.

The paper suggests there are operational hazards for both low flow conditions and for high flow conditions caused by the bridge at Kaunus.   The paper should contain a summary of what observations have been made about difficulties experienced by vessels using the waterway during extremes-in-flow conditions.

(4)   Explanation needed for discrepancy between observed and calculated velocities

In Line 452  it is noted that at cross-sections p7. P8, and p9 the measured velocity is higher than the observed velocity.  A possible explanation is that the effective width of the piers increases beyond their physical width as flow increases due to separation of flow – if so this can be allowed for in the mathematical model.  In any case an expanded explanation of this discrepancy is needed .

Additional minor editing changes

Line 20             founds     funds       from the EU

Line 36             The application of   ArcGIS was applied  in the post-processing phase lets  to …

Line 44                  inland waterway transport  hydraulics

Line 182                courses   causes the   frequent fluctuation of the water level ……….steadily.

Reviewer 3 Report

This article introduces the research methods very clearly. The figures and tables are made clearly and of high quality. It is recommended to publish. Considering the  length of the current article, it is suggested to reduce the length of the article appropriately.

Round 2

Reviewer 2 Report

Review of revised paper 2139109 Application of physical and numerical modeling for determination of waterway safety under the bridge in Kaunas City, Lithuania.

The stated purpose of this study is to determine the water-surface profile that would exist in the Nemunas River in the vicinity of the Kaunas Railway Bridge under conditions of very low flow and under high flow conditions.   Waterlevel profiles under the bridge were analysed using a physical model and using a numerical model (HEC-RAS).

The flow profile under the bridge is determined by four factors: (1) the flowrate in the river; (2) the three dimensional form of the river channel, including cross-sectional form and longitudinal slope; (3) the hydraulic roughness of the wetted perimeter of the channel; (4) the boundary condition of water depth at the downstream end of the river section for the specific flowrate being analysed.

Both the physical model and the numerical model require as input item (4) the waterlevel  at the downstream end of the section under study at the flowrate being studied.  If this waterlevel is not known then the model results derived from an arbitrary choice of downstream waterlevel cannot be used to draw conclusions about the safety of navigation under the bridge.

The authors state in the revised paper in Line 430 “it was assumed that the direct measurements of water level in the analysed object are not available for calibration of any model, nether physical nor numerical.”  This statement is not correct.  As stated in the text and shown in Figure 4a the Gauge Station Nemunas-Kaunas is located downstream (about 5 km) of the reach being studied.  The water

level in the Nemunas River is continuously measured at the Gauge Station.  This available record of water levels at the Gauge Station, together with the set of measurements of  water depth and wetted cross-section area in the flowrate-measurement cross-section which are made during periodic flowrate measurements, are available.  From this data set, combined with the record of flowrate in the river, it is possible to determine both the water depth that would be present at the downstream end of the bridge-reach section at each flowrate being studied and the effective hydraulic roughness of the Nemunas River channel in the vicinity of the bridge.

Until the water depth data available at the Gauging Station has been obtained and the correct water depth at the downstream cross-section of the study reach has been determined and used in runs of both the physical model and the numerical model and the results reanalysed  the results currently obtained are not reliable and the paper is not satisfactory for publication.

As a bonus of this recalculation the analysis at the Gauge Station will provide checks on the hydraulic roughness of the channel section which can be compared to the roughness values cited in the current paper.th

It appears from the cover letter that there are results from a 1-D Hec-RAS model that should provide both guidance on hydraulic roughness and estimates of downstream cross-section water depth for various flowrates.

Round 3

Reviewer 2 Report

Second Review of revised paper 2139109 Application of physical and numerical modeling for determination of waterway safety under the bridge in Kaunas City, Lithuania.

I specify below three issues that must be resolved to make the paper acceptable for publication

This paper is useful as an extension of paper by Szydlowski (2011- ref 27) on the topic of using numerical modelling to augment knowledge of the hydraulic properties of flow around bridge piers. In this paper flow properties of the channel of the Nemunas River in the vicinity of the Railway Bridge in Kaunas City Lithuania are modeled with an emphasis on distinguishing locations where safety hazards to navigation may exist.

A physical model of a  1.1 km reach of the Neumunas River was created. Flow depth profiles in the model were measured for three flowrates which represented  Q95% , Q5%, and Q1%    in the Nemunas River.  Numerical modelling using HEC-RAS was then conducted to confirm and augment the model results.

Major Issue One to be resolved

 In the physical and the numerical models one of the essential conditions that must be matched between the model and the actual situation in the Neumunas River is the waterlevel that is present at the downstream end of the modelled reach during flow at the specified rate used in the model run.

The authors state that no information is available to them on the observed flow depth in the downstream cross-section of the Nemunas River at the three flowrates used in this study.  In the absence of observed flow depths, a best estimate of flow depth must be made to be set in the physical model by control of the tailgate and in the numerical model by specified boundary condition at the downstream cross-section.

The best estimate of the relationship of flow depth versus flowrate at the downstream cross-section is to assume that a Normal flow depth applies. Normal flow depth is calculated by calculating a depth which produces a product of V*A equal to the design flowrate using the Manning Equation to determine the function of V versus d and the average channel width downstream of the final cross-section  to determine A versus d.

The calculation of the Normal depth for each of the three design flowrates requires an estimate of channel roughness and channel slope. Channel slope is stated to be 0.03 %.  Channel roughness is best estimated from a set of measurements of flowrate and cross-section area in the channel such as are made at a gauging station.   Other less reliable estimates of Manning’s n are available in the literature.

A major defect in the paper is the absence of information as to how the downstream waterlevel was determined and set for the physical and for the numerical model.  For the physical model it is stated on Line 415 that “the free flow condition is used in the outlet to determine the flow conditions in the waterway”.  This phrasing suggests that the water depth at the downstream cross-section was not a set boundary condition but rather was determined by the slope and roughness of the physical model which is not how the boundary condition should be set.

In contrast to the above statement, it is stated on Lines 347-348 that “In experiments with greater discharges, Q5% ,and Q1%, the additional water head was imposed in the outlet of the model to make measurement easier”.   This statement suggests that in the physical model tailwater levels were controlled but in some arbitrary manner not related to the expected Normal Flow Depth in the river for the specified flowrate.

This issue of what tailwater levels were set in the physical model at the various flowrates must be resolved before the paper is acceptable for publication.   This must be done by clearly stating that tailwater levels in the model were set to expected water depths in the full-scale river for each flowrate tested. A Table must be included in the paper specifying the full-scale tailwater depth in the river at each of the three flowrates. The table should also specify what Manning’s n in the full-scale river channel would produce this depth.

Issue Two to be resolved

The statement in Line 351 “Such values are applied are applied to validate the model before the major computations”  does not convey any meaning to me.  It must be replaced by a statement of what  criteria for validation of the model were used and some examples of numerical values that confirmed validation.

Issue 3 to be resolved   - Change of emphasis in text to convey the main objective of the paper

The main emphasis of the paper should be on the use of numerical modeling to verify and augment results from physical models to determine the performance of river channels.

Needed Steps

1                    Change the title of the paper to Combining Physical and Numerical Modeling to Improve Assessment of Waterway Safety for Kaunus City Bridge Crossing Lithuania.

2                    Replace the first two sentences in the Abstract with these sentences Line 142 “The purpose of the presented research is to integrate physical modeling and numerical simulation supported by GIS techniques to determine the safety of inland navigation conditions in a river reach that includes a bridge crossing.  The analysed reach is located on the Nemunas River in the City of Kaunus Lithuania.”

3                    Remove Lines beginning at Line 56 “There are several efforts”……and ending at line 84 “ development of inland shipping”  - this material is not relevant to this paper.

4                    Shorten Section 2 to about ten lines.  Leave out description of the Nemunas River and its watershed and a description and history of the bridge. These are not relevant to this paper. Include a description of the study reach and its immediate neighbouring sections in terms of what properties of the channel can affect hydraulic conditions (channel roughness for example).
